# Umbilical Cord Wraps around a Newborn’s Legs like Ankle Shackles

**DOI:** 10.3390/diagnostics14040444

**Published:** 2024-02-17

**Authors:** Kun-Long Huang, Ching-Chang Tsai, Hsin-Hsin Cheng, Yun-Ju Lai, Pei-Fang Lee, Te-Yao Hsu

**Affiliations:** 1Department of Obstetrics and Gynecology, Kaohsiung Chang Gung Memorial Hospital, Chang Gung University Collage of Medicine, Kaohsiung 833401, Taiwan; mr9221@cgmh.org.tw (K.-L.H.); aniki@cgmh.org.tw (C.-C.T.); chokovarous@cgmh.org.tw (H.-H.C.); lusionbear@cgmh.org.tw (Y.-J.L.); pf7938@cgmh.org.tw (P.-F.L.); 2Department of Obstetrics and Gynecology, Jiannren Hospital, Kaohsiung 811020, Taiwan; 3Department of Obstetrics and Gynecology, Fangliao General Hospital, Pingtung 940004, Taiwan

**Keywords:** umbilical cord entanglement, strangulation, marked fetal movement, maternal anxiety

## Abstract

A 36-year-old woman, gravida 3, para 1 (previous cesarean section), with one medical abortion, and no history of systemic diseases presented an unremarkable medical history during prenatal visits. The final prenatal ultrasound at 38 weeks of gestation showed a vertex presentation, a weight of 2600 g, a normal amniotic fluid level, and the placenta located on the posterior wall of the uterus. Fetal cardiotocography conducted before delivery reported a reactive heart rate without decelerations. The Doppler wave analysis of the fetal umbilical artery was normal (the ratio of peak-systolic flow velocity to the end-diastolic flow velocity was 2.5) without the absence or reversal of end-diastolic velocity. The total score of the fetal biophysical profile by ultrasound was 8. The night before the scheduled cesarean section, she experienced heightened anxiety and was unable to sleep, noting “crazy” fetal movements throughout the night. During the cesarean section, it was discovered that the umbilical cord was wrapped around the newborn’s legs, resembling ankle shackles. The fetal weight was 2740 g, and Apgar scores were 9 at the first minute and 10 at the fifth minute. The motility of the neonatal legs was normal without cyanosis or neurological adverse outcomes.

The entanglement of the umbilical cord (EUC) around a newborn’s neck, body, or trunk is a common phenomenon observed during delivery. Single loops, nuchal locations, and loose wrapping of the umbilical cord are frequently observed [1]. The adverse effects of perinatal outcomes resulting from tightly wrapped umbilical cords include decelerations in fetal heart rate and perinatal acidemia [2,3,4]. Abnormal umbilical cord lengths (<45 cm or >95 cm) are associated with operative intervention and intrapartum complications [5]. Identifying cases with EUC via prenatal ultrasound poses numerous challenges because the spontaneous reduction of the umbilical cord may occur in cases with simple EUC (less than 3 loops). The “divot sign,” which appears as multiple holes resembling a honeycomb around the fetal neck or body, can be observed in real-time ultrasonography. Power Doppler applications offer greater accuracy in sensitivity and specificity for diagnosing nuchal cords, as they depict dynamic images showing colored blood flow within the looped umbilical cords [6]. The international practice guidelines suggest documenting the number of vessels in the umbilical cord and the locations of bilateral insertion sites [7]. Even when the wrapping of the umbilical cord around the neck, trunk, or limbs is observed prenatally, delivery route practice is not altered if there is adequate amniotic fluid volume and a reactive fetal heart rate. However, a non-reassuring fetal heart rate, fetal growth restriction, and stillbirth were reported complications of the EUC [4,6]. Delivery in prematurity for those with EUC is not recommended in singletons, but antenatal steroids can be considered if signs of fetal distress are observed before delivery. Sherer et al. recommended daily assessment of fetal movement and twice-weekly fetal cardiotocography for gestational ages less than 37 weeks [6]. Only for monoamniotic twins with EUC, an earlier delivery at 32~34 weeks of gestation is reasonable due to higher mortality rates [8,9]. We present a rare and interesting case delivered by cesarean section, where the newborn’s legs were strangulated by the umbilical cord, resembling ankle shackles (Figure 1). The length of the umbilical cord (60 cm) and its insertion site on the placenta were normal. The motility and development of the legs was normal without cyanosis or neurological adverse outcomes at delivery and at two months old (Figure 2). Heazell et al. summarized the current evidence regarding potential causes of increased fetal movement, including infections, noxious stimuli, maternal anxiety, umbilical cord problems, and fetal seizures [10]. Despite neonatal outcomes being good without any consequences, the authors suggest that further examination is warranted to exclude ominous conditions such as fetal distress due to prolonged cord compression. Umbilical cord artery pH is a useful biomarker that provides more informative data to determine possible causes after birth [11].

## Figures and Tables

**Figure 1 diagnostics-14-00444-f001:**
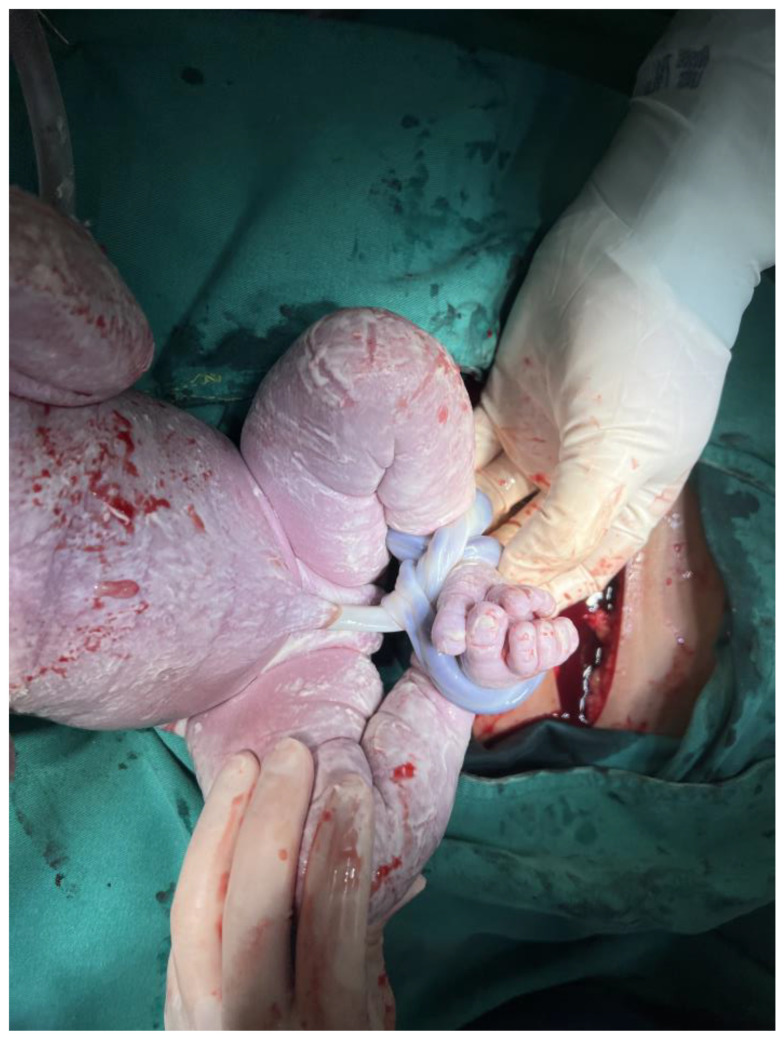
During the cesarean section, it was discovered that the umbilical cord was wrapped around the newborn’s legs, resembling ankle shackles. The umbilical cord encircled the left leg twice and the right leg once, was tightly wound and unable to be untangled spontaneously.

**Figure 2 diagnostics-14-00444-f002:**
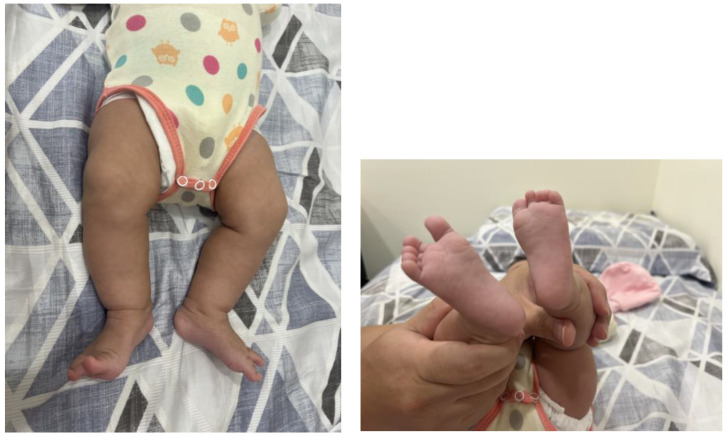
The motility and development of the neonatal legs was normal without cyanosis or neurological adverse outcomes at two months old.

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
