# Peer review of "Umbilical Cord Wraps around a Newborn’s Legs like Ankle Shackles"

_diagnostics, 2024, doi:10.3390/diagnostics14040444_

Round 1
Reviewer 1 Report
Comments and Suggestions for Authors
This is an interesting case. However, there are no ultrasounds to “confirm” prenatal diagnosis. You have right saying that even when the wrapping of the umbilical cord was observed prenatally, you did not alter your delivery route practice. However, He serial ultrasound examinations, including Doppler studies and biophysical profile scoring could estimate how “risky” is a particular case. Please make some comments about that.
Comments on the Quality of English LanguagePlease change "observed.(1)." to "observed (1)." and correct similar mistakes
Author Response
Thanks for your patiently reviewing our manuscript and the followings are our replies point by point:
- You have right saying that even when the wrapping of the umbilical cord was observed prenatally, you did not alter your delivery route practice. However, the serial ultrasound examinations, including Doppler studies and biophysical profile scoring could estimate how “risky” is a particular case. Please make some comments about that.
Ans: The Doppler wave analysis of the fetal umbilical artery at 38 weeks of gestation was normal (the ratio of peak systolic flow velocity to the end-diastolic flow velocity was 2.5) without the absence or reversal of end-diastolic velocity. The total score of the fetal biophysical profile by ultrasound was 8. Due to the absence of sign of fetal distress, we did not change our practice. The revision was made in line 53-55.
- Please change "observed.(1)." to "observed (1)." and correct similar mistakes.
Ans: We had revised the mistakes near the reference numbers.
- The English grammar was rechecked using Chat-GPT 3.5 website version.
Reviewer 2 Report
Comments and Suggestions for Authors
The Authors present an unusual image of a baby with cord entanglement around the ankles. However, the paper is too short and this interesting findings is not properly presented. The Authors should provide a brief introduction about cord entanglements, their prenatal diagnosis when possible, the main consequences and the countermeasures which can be taken. The Authors should expand the case report with more comments: how were the legs? were there any neurological and/or vascular consequences for the legs and feet? how was their motility? were there any prenatal images? was this condition at least suspected, even after the C-section? The Authors should also provide a brief discussion regarding their findings, if previously reported in literature, the main consequences etc...
Comments on the Quality of English LanguageModerate English editing is required.
Author Response
Thanks for your patiently reviewing our manuscript and the followings are our replies point by point:
- The Authors present an unusual image of a baby with cord entanglement around the ankles. However, the paper is too short and this interesting findings is not properly presented. The Authors should provide a brief introduction about cord entanglements, their prenatal diagnosis when possible, the main consequences and the countermeasures which can be taken.
Ans: We had added a brief introduction regarding prenatal diagnosis through ultrasound, associated consequences, and countermeasures in lines 18-24 and 27-33.
- The Authors should expand the case report with more comments: how were the legs? were there any neurological and/or vascular consequences for the legs and feet? how was their motility? were there any prenatal images? was this condition at least suspected, even after the C-section?
Ans: The motility of the neonatal legs was normal without cyanosis or neurological adverse outcomes at delivery and at two months old (revision in line 35-36 and add figure 2, line 63-65). Unfortunately, we did not record the ultrasound images of both legs before cesarean section because we did not consider the possible diagnosis of "cord entanglement on the legs".
- The Authors should also provide a brief discussion regarding their findings, if previously reported in literature, the main consequences etc...
Ans: We had added a brief discussion and summary from the most similar literature in lines 36-42.
- The English grammar was rechecked using Chat-GPT 3.5 website version.
Round 2
Reviewer 2 Report
Comments and Suggestions for Authors
The Abstract should be composed of a maximum of 200 words and must not include references. Please provide a separate introduction or add the relevant information to the images.
Comments on the Quality of English LanguageModerate English editing is required.
Author Response
For reviewer 2:
Thanks for your patiently reviewing our revised manuscript and the followings are our replies point by point:
- The Abstract should be composed of a maximum of 200 words and must not include references. Please provide a separate introduction or add the relevant information to the images.
Ans: The abstract was shortened to 198 words without reference. We also revised the article structure and added relevant information to describe the images (highlighted with the yellow color).
- The English grammar was rechecked using Chat-GPT 3.5 website version.